# The independent factors associated with oxygen therapy in COVID-19 patients under 65 years old

**Yue-Nan Ni[o], Ting Wang[o], Bin-miao Liang\*, Zong-An Liang[o]\***

Department of Respiratory and Critical Care, West China School of Medicine and West China Hospital, Sichuan University, Chengdu, China

[o] These authors contributed equally to this work.
\* liangbinmiao@163.com (BL); liangzatg@126.com (ZL)

## Abstract

### Background

The number of hospitalized young coronavirus disease 2019 (COVID-19) patients has increased significantly. However, specific data about COVID-19 patients under 65 years old who are admitted to the hospital are scarce.

### Methods

The COVID-19 patients under 65 years old who were admitted to the hospital in Sichuan Province, Renmin Hospital of Wuhan University, and Wuhan Red Cross Hospital were included in this study. Demographic information, laboratory data and clinical treatment courses were extracted from electronic medical records. Risk factors associated with oxygen therapy were explored.

### Results

Eight hundred thirty-three COVID-19 patients under 65 years old were included. Of the included patients, 29.4% had one or more comorbidities. Oxygen therapy was required in 63.1% of these patients, and the mortality was 2.9% among the oxygen therapy patients. Fever (odds ratio [OR] 2.072, 95% confidence interval [CI] 1.312–3.271, $p = 0.002$), dyspnea (OR 2.522, 95% CI 1.213–5.243, $p = 0.013$), chest distress (OR 2.278, 95% CI 1.160–4.473, $p = 0.017$), elevated respiratory rate (OR 1.114, 95% CI 1.010–1.228, $p = 0.031$), and decreased albumin (OR 0.932, 95% CI 0.880–0.987, $p = 0.016$) and globulin levels (OR 0.929, 95% 0.881–0.980, $p = 0.007$) were independent factors related to oxygen therapy.

### Conclusions

Oxygen therapy is highly required in COVID-19 patients under 65 years old who are admitted to the hospital, but the success rate is high. Respiratory failure-related symptoms, elevated respiratory rate, low albumin and globulin levels, and fever at admission are independent risk factors related to the requirement of oxygen.

**Data Availability Statement:** All relevant data are within the manuscript and Supporting Information files.

**Funding:** This study was partly supported by the National Key Research and Development Program

of China (2016YFC1304303), awarded to Zong-An Liang, and the Sichuan Science and Technology Agency Grant (2019YFS0033), awarded to Bin-Miao Liang and Sichuan Science and Technology Project (2020YFS0005), awarded to Ting Wang. The funders had no role in study design, data collection and analysis, decision to publish, or preparation of the manuscript.

**Competing interests:** None of all authors have any financial or non-financial competing interests in this manuscript.

**Abbreviations:** ALT, alanine aminotransferase; APTT, activated partial thromboplastin time; ARDS, acute respiratory distress syndrome; AST, aspartate aminotransferase; CI, confidence interval; COPD, chronic obstructive pulmonary distress; COVID-19, novel coronavirus disease 2019; ECMO, extracorporeal membrane oxygenation; ESR, erythrocyte sedimentation rate; ICU, intensive care unit; IQR, interquartile range; LOS, length of stay; MERS, Middle East respiratory syndrome; OR, odds ratio; PT, prothrombin time; RR, respiratory rate.

## Introduction

The novel coronavirus disease 2019 (COVID-19) is caused by a virus named SARS-CoV-2. The major symptoms included fever, cough and dyspnea [1, 2]. The minor symptoms are alteration of smell and taste, gastrointestinal symptoms, headache, and cutaneous manifestations [3–5]. It has spread to more than 200 countries and caused more than one million deaths as of 28th Sep, 2020. Approximately 29% of infected patients are admitted to the hospital [6]. According to a report from New York, 27.8% of the patients who were admitted to the hospital required oxygen therapy, and 12% of hospitalized patients received invasive mechanical ventilation, among which 88% died [7]. Eighty-six percent of the patients who died had complications related to respiratory failure [8]. Studies have shown that hypoxemia is independently associated with in-hospital mortality [9]. Thus, administrating oxygen in time is essential.

Although the mortality in young patients remains low [10], the proportion of young people among those admitted to the hospital is not lower than that of older people (45.6% vs. 54.4%) [6]. Studies have shown that the need for ventilators, intensive care unit (ICU) admission and fatality were significantly higher in COVID-19 patients older than 65 years old than in young patients [11, 12]. However, no specific data about the clinical characteristics and treatment of young COVID-19 patients admitted to the hospital have been shown.

Since many countries have reopened, a second spike in COVID-19 cases has been triggered. It was reported that cases of young COVID-19 patients increased significantly [13]. Thus, understanding the severity of COVID-19 patients under 65 years old who need hospitalization helps estimate the medical burden. Moreover, exploring the risk factors related to the requirement of oxygen therapy helps triage patients and improve clinical outcomes. Thus, we conducted this retrospective study to determine the severity of COVID-19 patients under 65 years old, the requirement for medical resources and related factors.

## Methods

All the patients infected with SARS-CoV-2 who were admitted to the hospital in Sichuan Province or admitted to Renmin Hospital of Wuhan University or Wuhan Red Cross Hospital during 16th Jan 2020 and 30th March were retrospectively screened. The inclusion criteria were as follows: 1) patients were diagnosed with COVID-19 according to the World Health Organization interim guidance [14]: nasal and/or pharyngeal samples were obtained from all patients at admission and tested using real-time reverse transcriptase–polymerase chain reaction; 2) patients were between 18 and 65 years old. The ethics committee of Sichuan University, West China Hospital, Renmin Hospital of Wuhan University and Wuhan Red Cross Hospital approved this study. Because this was a retrospective observational study that included no therapeutic intervention, written informed consent was waived.

From the electronic medical records, we extracted the demographic data, clinical symptoms and signs at admission, comorbidities (including chronic heart disease, chronic obstructive pulmonary disease (COPD), chronic renal disease, obesity, diabetes, hypertension, immunodeficiency and chronic liver disease), the results of laboratory tests at admission (including a complete blood count, blood chemical analysis, coagulation testing, assessment of liver and renal function, etc.), treatment details, complications and clinical outcome. Fever was defined as body temperature equal to or higher than 37.5°C.

The primary outcome was the need for oxygen therapy, which was conducted when emergency signs (obstructed or absent breathing, severe respiratory distress, central cyanosis, shock, coma, or convulsions) were observed, following the guidelines [14]. Secondary outcomes included noninvasive ventilation, invasive ventilation, ICU admission, complications and hospital mortality.

## Statistical analysis

Continuous variables were reported as the mean±standard deviation if the variable had a standard normal distribution or the median (interquartile range [IQR]) if not. The Z test was used to test whether the variable had a standard normal distribution. Categorical variables were reported as frequency and proportion. Student's t-test, the Mann-Whitney U-test and the Kruskal-Wallis test were used for comparisons between continuous variables, and the Chi-squared test or Fisher's exact test was used for comparisons between categorical variables, as appropriate.

To explore the factors related to oxygen therapy, we constructed a logistic regression model. Variables with a $p$ value <0.1 in the univariate analysis were entered in a multivariate logistic regression analysis to identify independent risk factors associated with oxygen therapy.

The analyses regarding different factors were based on nonmissing data. Missing data were not imputed. All the analyses were performed with SPSS 19.0 (SPSS Inc.), and a 2-sided p-value less than 0.05 was considered statistically significant.

# Results

## Demographic and clinical characteristics

Eight hundred thirty-three patients with COVID-19 between 18 and 65 years old were included in our study. A total of 526 (63.1%) patients needed oxygen therapy. The median age was 48 (37–56) years in patients who needed oxygen therapy and 40 (30–51) years in those who did not ($p$<0.001) (Table 1). The proportion of males was 50.4% and 44% ($p$ = 0.020), respectively. No significant difference was found in smoking status (current smoker: 9.3% vs. 12.7%, $p$ = 0.125; former smoker 4.4% vs. 3.9%, $p$ = 0.748) between the patients who needed oxygen therapy and those who did not. The proportion of current alcohol abuse patients was not significantly different between the two groups (9.7% vs. 10.7%, $p$ = 0.626). The most common coexisting condition was hypertension, 17.1% in the oxygen therapy group and 11.1% in the other group, $p$ = 0.018. A total of 245 (29.4%) patients had coexisting conditions. Seventy-seven patients had at least 2 coexisting conditions. No significant difference was found in other coexisting conditions between the two groups.

## Symptoms and signs at admission

In the oxygen therapy group, more patients had fever (70.9% vs. 55%, $p$<0.001), dry cough (59.5% vs. 41%, $p$<0.001), productive cough (35% vs. 22.5%, $p$<0.001), dyspnea (23.6% vs. 6.8%, $p$<0.001), fatigue (18.3% vs. 10.7%, p = 0.004), wheezing (9.5% vs. 4.6%, $p$ = 0.010), chest distress (21.9% vs. 7.8%, $p$<0.001), and diarrhea (12% vs. 6.2%, $p$ = 0.007) at admission. A higher respiratory rate (RR) of 20 (19–21) vs. 20 (19–20) times/minute ($p$ = 0.005) was found in oxygen therapy patients, and more patients had a RR >20 (30.7% vs. 19.5%, $p$<0.001) in the oxygen therapy group.

## Laboratory indices

In the patients with oxygen therapy, higher white blood cell counts (6.09 ± 3.78 vs. 5.54 ± 1.95 ×$10^9$/mL, $p$ = 0.009), lower level of hemoglobin (131.17 ± 22.75 vs. 135.20 ± 22.70 g/L, $p$ = 0.021), lower percentage of lymphocyte (23.82 ± 12.05 vs. 27.07 ± 9.99 $p$<0.001), higher neutrophil% (65.73 ± 16.37 vs. 61.49 ± 13.50 ×$10^9$/L, $p$<0.001), higher neutrophil/lymphocyte ratio (2.80 (1.83–4.87) vs. 2.31 (1.48–3.61), $p$<0.001) and higher value of glucose (5.63(4.92–7.10) vs. 5.40(4.76–6.53) mmol/L, $p$ = 0.018) were found. Meantime, in patients who received oxygen therapy, both the levels of alanine aminotransferase (ALT) (27 (17–42.25) vs. 23 (15–

**Table 1. The characteristics of patients received oxygen therapy and not.**

| | All patients (n = 833) | Oxygen therapy(n = 526) | Non-oxygen therapy(n = 307) | p |
|---|---|---|---|---|
| Age, year[b] | 46(34–55) | 48(37–56) | 40(30–51) | <0.001 |
| Male, no./total no. (%) | 400/833(48.0%) | 265/526 (50.4%) | 135/307 (44%) | 0.020 |
| Current smoker, no./total no. (%) | 88/833 (10.6%) | 49/526 (9.3%) | 39/307 (12.7%) | 0.125 |
| Former smoker, no./total no. (%) | 35/833 (4.2%) | 23/526 (4.4%) | 12/307 (3.9%) | 0.748 |
| Current alchohol, no./total no. (%) | 84/833 (10.08%) | 51/526 (9.7%) | 33/307 (10.7%) | 0.626 |
| **Symptom at admission** | | | | |
| Fever, no./total no. (%) | 542/833 (65.1%) | 373/526 (70.9%) | 169/307 (55%) | <0.001 |
| Dry cough, no./total no. (%) | 439/833 (52.7%) | 313/526 (59.5%) | 126/307 (41%) | <0.001 |
| Productive cough, no./total no. (%) | 253/833 (30.4%) | 184/526 (35%) | 69/307 (22.5%) | <0.001 |
| Dyspnea, no./total no. (%) | 145/833 (17.4%) | 124/526 (23.6%) | 21/307 (6.8%) | <0.001 |
| Sore throat, no./total no. (%) | 263/833 (31.6%) | 178/526 (33.8%) | 85/307 (27.7%) | 0.065 |
| Fatigue, no./total no. (%) | 129/833 (15.5%) | 96/526 (18.3%) | 33/307 (10.7%) | 0.004 |
| Wheezing, no./total no. (%) | 64/833 (7.7%) | 50/526 (9.5%) | 14/307 (4.6%) | 0.010 |
| Chest distress, no./total no. (%) | 139/833 (16.7%) | 115/526 (21.9%) | 24/307 (7.8%) | <0.001 |
| Myalgia, no./total no. (%) | 87/833 (10.4%) | 58/526 (11%) | 29/307 (9.4%) | 0.472 |
| Headache, no./total no. (%) | 63/833 (7.6%) | 36/526 (6.8%) | 27/307 (8.8%) | 0.298 |
| Vomiting, no./total no. (%) | 33/833 (4.0%) | 24/526 (4.6%) | 9/307 (2.9%) | 0.244 |
| Diarrhea, no./total no. (%) | 82/833 (9.8%) | 63/526 (12%) | 19/307 (6.2%) | 0.007 |
| Body temperature,°C[b] | 36.7(36.5–37.2) | 36.8(36.5–37.3) | 36.70(36.5–37.1) | 0.146 |
| Heart rate, times/minute[a] | 88.77±25.03 | 89.39± 29.67 | 87.70 ±13.82 | 0.350 |
| RR, times/minute | 20(19–21) | 20(19–21) | 20(19–20) | 0.005 |
| RR>20, no./total no. (%) | 220/833 (26.4%) | 160/526 (30.7%) | 60/307 (19.5%) | <0.001 |
| **Comorbidities** | | | | |
| Heart disease, no./total no. (%) | 31/833 (3.7%) | 24/526 (4.6%) | 7/307 (2.3%) | 0.093 |
| COPD, no./total no. (%) | 9/833 (1.0%) | 7/526 (1.3%) | 2/307 (0.7%) | 0.498 |
| CKD, no./total no. (%) | 11/833 (1.3%) | 9/526 (1.7%) | 2/307 (0.7%) | 0.345 |
| Liver disease, no./total no. (%) | 63/833 (7.6%) | 35/526 (6.7%) | 28/307 (9.1%) | 0.194 |
| Obesity, no./total no. (%) | 4/833 (0.5%) | 1/526 (0.2%) | 3/307 (1%) | 0.144 |
| Diabetes, no./total no. (%) | 76/833 (9.1%) | 54/526 (10.3%) | 22/307 (7.2%) | 0.134 |
| Iimmunodeficiency, no./total no. (%) | 10/833 (1.2%) | 8/526 (1.5%) | 2/307 (0.7%) | 0.339 |
| Hypertension, no./total no. (%) | 124/833 (14.9%) | 90/526 (17.1%) | 34/307 (11.1%) | 0.018 |
| Number of comorbidities | | | | |
| 1, no./total no. (%) | 168/833 (20.2%) | 113/526 (21.5%) | 55/307 (17.9%) | 0.216 |
| 2, no./total no. (%) | 66/833 (7.9%) | 47/526 (9%) | 19/307 (6.2%) | 0.157 |
| ≥3, no./total no. (%) | 11/833 (1.3%) | 7/526 (1.4%) | 4/307 (1.3%) | 0.973 |
| **Laboratory indices** | | | | |
| White blood cells, ×10⁹/mL[a] | 5.90±3.26 | 6.09±3.78 | 5.54±1.95 | 0.009 |
| Hemoglobin, g/L[a] | 132.61±22.80 | 131.17±22.75 | 135.20±22.70 | 0.021 |
| Platelets count×10⁹/L[a] | 216.10±88.34 | 214.74±86.44 | 218.6±291.88 | 0.573 |
| Lymphocytes,%[a] | 24.97±1.46 | 23.82±12.05 | 27.07±9.99 | <0.001 |
| Neutrophils, %[a] | 64.23±15.54 | 65.73± 16.37 | 61.49 ±13.50 | <0.001 |
| Neutrophils / Lymphocytes ratio[b] | 2.61(1.69–4.21) | 2.80(1.83–4.87) | 2.31(1.48–3.61) | <0.001 |
| D dimer, mg/L[b] | 0.48(0.24–0.96) | 0.38(0.19–0.81) | 0.56(0.31–1.06) | 0.001 |
| Total bilirubin, μmol/L[b] | 10.05(7.2–14.05) | 10.00(6.95–14.20) | 10.10(7.30–14.00) | 0.546 |
| ALT, U/L[b] | 26(16–41) | 27(17–42.25) | 23(15–39) | 0.008 |
| AST, U/L[b] | 24(19–35) | 25(20–36) | 23.9(19–33) | 0.045 |

(*Continued*)

**Table 1.** (Continued)

| | All patients (n = 833) | Oxygen therapy(n = 526) | Non-oxygen therapy(n = 307) | *p* |
|---|---|---|---|---|
| AST/ALT[b] | 1.00(0.75–1.32) | 1.02(0.76–1.37) | 1.00(0.73–1.29) | 0.317 |
| Total protein, g/L[a] | 67.36±7.97 | 65.92 ±7.10 | 69.87 ±8.76 | <0.001 |
| Albumin, g/L[a] | 40.44±5.82 | 39.33± 6.09 | 42.38 ±4.73 | <0.001 |
| Globulin, g/L[a] | 26.84±5.38 | 26.31± 5.12 | 27.75 ±5.69 | <0.001 |
| Urea, mmol/L [a] | 3.81(3.10–5.01) | 3.81(3.19–4.73) | 3.81(3.09–5.16) | 0.774 |
| Creatinine, umol/L [a] | 65.50±42.91 | 66.43± 52.39 | 63.87± 16.63 | 0.475 |
| Glucose, mmol/L [b] | 5.54(4.84–6.93) | 5.63(4.92–7.10) | 5.40(4.76–6.53) | 0.018 |
| APTT, s[a] | 29.93±5.64 | 29.92 ±5.76 | 30.07 ±5.11 | 0.754 |
| PT, s[a] | 12.54±3.54 | 12.68 ± 4.25 | 12.30 ±1.61 | 0.225 |
| Hospital length of stay, d[b] | 15(9–22) | 16(10–24) | 14.5(10–20) | 0.002 |

Body temperature is missing in 18 patients, heart rate is missing in 8 patients, respiratory rate is missing in 9 patients, white blood cells is missing in 93 patients, hemoglobin is missing in 98 patients, platelets is missing in 108 patients, lymphocytes is missing in 91 patients, neutrophils is missing in 92 patients, neutrophils / lymphocytes is missing in 93 patients, monocytes 97 is missing, D dimer is missing in 315 patients, total bilirubin is reported in 618 patients, ALT is missing in 226 patients, AST is missing in 236 patients, total protein is missing in 210 patients, albumin is missing in 208 patients, globulin is missing in 210 patients, urea is missing in 246 patients, creatinine is missing in 243 patients, glucose is missing in 237 patients, APTT is missing in 283 patients, PT is missing in 285 patients.

ALT, alanine aminotransferase; APTT, activated partial thromboplastin time; AST, aspartate aminotransferase; CKD, chronic kidney disease; COPD, chronic obstructive pulmonary disease; PT, prothrombin time; RR, respiratory rate; T, temperature

[a]mean±SD

[b]median(IQR)

39), U/L *p* = 0.008) and the levels of aspartate aminotransferase (AST) levels (25 (20–36) vs. 23.9 (19–33) U/L, *p* = 0.045) were higher. In the oxygen therapy group, lower levels of total protein (65.92 ±7.10 vs. 69.87 ±8.76 g/L, p<0.001), albumin (39.33± 6.09 vs. 42.38 ±4.73 g/L, *p*<0.001) and globulin (26.31± 5.12 vs. 27.75 ±5.69 g/L, *p*<0.001) and D-dimer of (0.38(0.19–0.81) vs. 0.56(0.31–1.06) mg/L, p = 0.001) were found. No significant differences in total bilirubin, the ratio of AST/ALT, creatinine, urea, activated partial thromboplastin time (APTT), or prothrombin time (PT) were found between the two groups.

## Risk factors related to oxygen therapy

After adjustment, we found that patients with fever (odds ratio [OR] 2.072, 95% confidence interval [CI] 1.312–3.271, *p* = 0.002) and/or dyspnea (OR 2.522, 95% CI 1.213–5.243, *p* = 0.013) and chest distress (OR 2.278, 95% CI 1.160–4.473, *p* = 0.017) at admission had an increased risk of oxygen therapy. A higher respiratory rate (OR 1.114, 95% CI 1.010–1.228, *p* = 0.031) and lower albumin (OR 0.932, 95% CI 0.880–0.987, *p* = 0.016) and globulin levels (OR 0.929, 95% 0.881–0.980, *p* = 0.007) were also associated with a higher risk of oxygen therapy (Table 2).

## Treatment and complications of patients with oxygen therapy

Among all the patients who needed oxygen therapy, seven patients received (1.3%) high-flow nasal cannula intervention, and 41 (7.8%) patients received noninvasive ventilation. Twenty-six (4.9%) patients were admitted to the ICU, 7 (1.3%) patients needed invasive ventilation, and two (0.4%) patients received extracorporeal membrane oxygenation (ECMO). The length of hospitalization was longer in patients who received oxygen therapy than in those who did not (16.00(10–24) vs. 14.50(10–20) days, *p* = 0.002). Twenty-six (4.9%) patients developed

**Table 2. Logistic regression analysis of factors associated with oxygen therapy.**

| | Univariable logistic regression, OR(95%CI) | p | Multivariable logistic regression, OR(95%CI) | p |
|---|---|---|---|---|
| Age, year | 1.040(1.027–1.052) | <0.001 | 1.015(0.995–1.035) | 0.137 |
| Male, yes or no | 0.773(0.583–1.026) | 0.074 | 0.836(0.518–1.348) | 0.462 |
| Fever, yes or no | 1.991(1.485–2.669) | <0.001 | 2.072(1.312–3.271) | 0.002 |
| Drycough, yes or no | 2.111(1.585–2.811) | <0.001 | 1.539(0.995–2.380) | 0.053 |
| Productive cough, yes or no | 1.856(1.344–2.562) | <0.001 | 1.077(0.669–1.732) | 0.761 |
| Dyspnea, yes or no | 4.201(2.582–6.835) | <0.001 | 2.522(1.213–5.243) | 0.013 |
| Fatigue, yes or no | 0.493(0.232–1.051) | 0.067 | 0.469(0.167–1.315) | 0.150 |
| Sore throat, yes or no | 1.336(0.981–1.819) | 0.066 | 0.722(0.454–1.147) | 0.167 |
| Chest distress, yes or no | 3.299(2.072–5.254) | <0.001 | 2.278(1.160–4.473) | 0.017 |
| Wheezing, yes or no | 2.198(1.194–4.407) | 0.011 | 0.961(0.319–2.892) | 0.943 |
| Diarrhea, yes or no | 2.063(1.210–3.517) | 0.008 | 1.395(0.680–2.862) | 0.364 |
| Body temperature,°C | 1.233(1.007–1.509) | 0.043 | 1.114(0.802–1.550) | 0.519 |
| RR, times/minute | 1.133(1.065–1.204) | <0.001 | 1.114(1.010–1.228) | 0.031 |
| Hypertension, yes or no | 1.657(1.086–2.529) | 0.019 | 0.728(0.401–1.322) | 0.297 |
| Total protein, g/L | 0.930(0.908–0.953) | <0.001 | 1.009(0.969–1.051) | 0.668 |
| Albumin, g/L | 0.888(0.856–0.922) | <0.001 | 0.932(0.880–0.987) | 0.016 |
| Globulin, g/L | 0.952(0.923–0.981) | 0.001 | 0.929(0.881–0.980) | 0.007 |
| White blood cell count, $\times 10^9$/L | 1.074(1.008–1.144) | 0.028 | 1.054(0.952–1.166) | 0.311 |
| Hemoglobin, g/L | 0.992(0.985–0.999) | 0.026 | 0.994(0.984–1.004) | 0.272 |
| Lymphocytes, % | 0.975(0.962–0.989) | <0.001 | 1.008(0.973–1.043) | 0.667 |
| Neutrophils cell, % | 1.018(1.008–1.028) | <0.001 | 1.007(0.981–1.033) | 0.603 |
| Neutrophils/ Lymphocytes ratio | 1.144(1.077–1.215) | <0.001 | 1.099(0.951–1.269) | 0.201 |
| ALT, U/L | 1.011(1.004–1.018) | 0.003 | 1.009(0.998–1.021) | 0.118 |
| AST, U/L | 1.011(1.001–1.021) | 0.027 | 0.993(0.982–1.005) | 0.239 |

ALT, alanine aminotransferase; APTT, activated partial thromboplastin time; AST, aspartate aminotransferase; CI, confidence interval; OR, odds ratio; PT, prothrombin time; RR, respiratory rate

The items with p value <0.1 in the univariable logistic regression analysis were included in the multivariable logistic regression analysis.

acute respiratory distress syndrome (ARDS), and three (0.6%) patients developed myocarditis. Arrhythmia occurred in ten (1.9%) patients. Gastrointestinal bleeding was found in three (0.6%) patients, and liver dysfunction was found in 74 (14%) patients. The mortality rate among patients who received oxygen therapy was 2.9%. No patients died among those who never received oxygen therapy. The details are listed in Table 3.

## Discussion

Our study found that oxygen therapy is highly required in hospitalized COVID-19 patients under 65 years old. The success rate of oxygen therapy is high, and the risk of developing complications is low. Fever, dyspnea, chest distress, respiratory rate, and albumin and globulin levels at admission are independent factors associated with the requirement of oxygen therapy.

Our study showed that approximately 63% of hospitalized COVID-19 patients under 65 years old need oxygen therapy; however, the failure rate of oxygen therapy is below 10%, and the mortality rate is only 2.9% among oxygen therapy patients. Additionally, the length of hospitalization was longer in the oxygen therapy group. This finding is reasonable since those that required oxygen were much more severe. However, the difference in the length of hospitalization was only approximately 1.5 days. Thus, it is indicated that once oxygen therapy is administered in time, most young patients have a good clinical outcome. Thus, triaging patients who

**Table 3. Treatment details and complications in 526 patients received oxygen therapy.**

|  | Oxygen therapy |
|---|---|
| HFNC, no./total no. (%) | 7/526 (1.3%) |
| Noninvasive ventilation, no./total no. (%) | 41/526 (7.8%) |
| ICU admission, no./total no. (%) | 26/526 (4.9%) |
| Invasive ventilation, no./total no. (%) | 7/526 (1.3%) |
| ECMO, no./total no. (%) | 2/526 (0.4%) |
| Complication, no./total no. (%) |  |
| ARDS, no./total no. (%) | 26/526 (4.9%) |
| Myocarditis, no./total no. (%) | 3/526 (0.6%) |
| Arrhythmia, no./total no. (%) | 10/526 (1.9%) |
| Gastrointestinal bleeding, no./total no. (%) | 3/526 (0.6%) |
| Liver dysfunction, no./total no. (%) | 74/526 (14%) |
| Dead, no./total no. (%) | 15/526 (2.9%) |

ARDS, acute respiratory distress syndrome, ECMO, extracorporeal membrane oxygenation; HFNC, high flow nasal cannula; ICU, intensive care unit

need oxygen therapy and administering oxygen therapy in time to avoid oxygen debt is very important to improve the clinical outcome. Moreover, the risk for COVID-19 patients under 65 years old to develop complications was very low. The most common complication in our study was liver impairment. Among our included patients, more than 90% received antivirals, more than half were administered antibiotics, and approximately 20% were treated with corticosteroids. All of these medicines cause liver impairment [15]. Furthermore, the hyperinflammation induced by COVID-19 could also damage liver function [16].

Dyspnea, chest distress and respiratory rate are independent factors associated with the requirement of oxygen therapy. All these symptoms were early signs of respiratory failure. Our study found that more patients had respiratory rates of more than 20 times/minute at admission in the oxygen therapy group (30.7% vs. 19.5%, p<0.001). Thus, respiratory rate and respiratory signs should be closely monitored. Moreover, low albumin and globulin levels were also independent risk factors. A similar result was also found in Middle East respiratory syndrome (MERS) patients [17]. It is believed that albumin reflects the baseline nutritional status of patients and is linked to poor clinical outcome for hospitalized patients [18]. Good nutrition could support the body with immunity to clear the virus and promote disease recovery [19]. Moreover, serum globulin is generated by the immune system and liver. It plays a vital role in host protection against infection. In MERS patients, the levels of IgG and neutralizing antibodies were weakly and inversely correlated with lower respiratory tract viral loads [20]. A previous study found that the antibody titer was independently associated with the severity of COVID-19 [21]. Moreover, after receiving convalescent plasma therapy, the antibody level in COVID-19 patients increased significantly, and the viral load decreased [22]. This finding indicated that globulin also plays a vital role in clearing SARS-CoV-2.

Moreover, our study found that fever is an independent factor associated with the requirement of oxygen therapy. However, the difference between the average body temperature in the two groups was actually very small. As mentioned in a previous study, fever might indicate a high risk of pulmonary function damage [23]. Additionally, fever was related to the upregulation of inflammatory cytokines, which might contribute to cytokine storms [24]. In contrast, upregulation of the inflammatory reaction helps viral clearance [25]. Thus, it is difficult to judge whether fever is beneficial to patients. Studies have shown that the duration of fever is

related to ICU admission [26]. We suppose that both the level of body temperature change and duration of fever should be taken into consideration.

Significant differences were found in the levels of white blood cells, lymphocytes, neutrophils and the ratio of neutrophils to lymphocytes. However, after adjustment, none of them was an independent risk factor. It has been reported that the levels of white blood cells and lymphocytes do not remain constantly higher or lower than normal values; it is a dynamic process [27]. The levels would be normal at first, followed by decreasing to a nadir after approximately a week, then increasing to a peak in the second week, and then decreasing again. Lymphocyte levels also decrease first and then increase. Moreover, the whole process could be divided into a defense-based protective phase and a second inflammation-driven damaging phase [28]. Both phases could cause severe infection and hospitalization, although most people would experience only the first phase. However, we could not determine which phase patients were in when they were admitted to the hospital according to the blood test results at one time point. Thus, we believe that white blood cells, lymphocytes or other inflammation-related biomarkers tested at one time point have difficulty reflecting the inflammation status and predicting the clinical outcome. Dynamic monitoring of the inflammation process might be better in predicting the prognosis. Moreover, the results of our study were different from those of previous studies [1, 29]. One reason was that there were some missing data in the laboratory results. Another reason was that our patients were under 65 years old, which was different from the situation in previous studies in which the included patients were much older.

Previous study found that the ratio of AST/ALT was a predictor for hospital mortality in patients with COVID-19 [30]. However, our study did not find any significant difference in the ratio of AST/ALT in patients with oxygen therapy and the ones without. Although the requirement of oxygen therapy and AST/ALT could be a marker of illness severity, other treatment process, such as hydroxychloroquine and corticosteroids [31, 32], might also influence the hospital mortality.

Previous studies showed that comorbidities, such as chronic obstructive pulmonary disease (COPD), diabetes, hypertension and malignancy, and the number of comorbidities were also related to poor prognosis in COVID-19 patients [33, 34]. However, in our study, no comorbidity was a risk factor for oxygen therapy after adjustment. On the one hand, the patients in our study were relatively young. The rates of comorbidity and of more than 1 comorbidity were lower than those in a previous report [11]. It is possible that our sample is too small to show statistical significance. On the other hand, we could not exclude the possibility that clinicians would suppose that young patients with combined coexisting conditions might have worse outcomes than those who did not, thus giving more attention to those patients. Studies have shown that mortality in critically ill patients is influenced by the patient-to-staff ratio [35]. However, no comorbidity was a risk factor for oxygen therapy, which does not mean that patients with coexisting conditions are safe. Instead, it indicates that the patients without coexisting condition should be given the same attention in case of exacerbation and the requirement of oxygen.

Moreover, smoking did not contribute to the outcome in the study, while a dose-response association between pack-years of cigarette smoking and pulmonary disease has been found [36]. Thus, we think the major explanation for the inconsistent result might be that the included patients in our study were relatively young and that cigarette smoking exposure was not significant enough to influence the clinical outcome in COVID-19 patients. Whereas a previous study suggested that smoke might be the major reason why male COVID-19 patients had poorer clinical outcomes [37], no sex difference was found in our study.

The limitations of our study should be addressed. First, due to the limitations of retrospective studies, the blood test data of some patients were missing. This fact would influence the

results and application of our study. Second, symptoms and coexisting conditions at admission were self-reported. Thus, there might be some inaccuracy. Last, the reason for oxygen therapy was difficult to define, although clinical practice guidelines were strictly followed by clinicians in all centers. Additionally, in the guidelines, the need for oxygen therapy is not indicated by arterial blood gas, perhaps due to the inconvenience of this special infectious disease. Thus, we could not determine whether every patient who received oxygen therapy had hypoxemia.

## Conclusions

Oxygen therapy is highly required and has a very high success rate in young patients with COVID-19. Early symptoms of respiratory failure, a higher respiratory rate, and lower albumin and globulin levels are independently associated with oxygen therapy.

## Supporting information

**S1 Data.**
(XLSX)

## Author Contributions

**Conceptualization:** Yue-Nan Ni.

**Data curation:** Ting Wang.

**Formal analysis:** Yue-Nan Ni.

**Funding acquisition:** Ting Wang, Bin-miao Liang, Zong-An Liang.

**Methodology:** Ting Wang, Bin-miao Liang.

**Resources:** Zong-An Liang.

**Supervision:** Bin-miao Liang, Zong-An Liang.

**Writing – original draft:** Yue-Nan Ni, Ting Wang.

**Writing – review & editing:** Bin-miao Liang, Zong-An Liang.

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
