## [Decision Letter · Decision Letter 0]

23 Sep 2020

PONE-D-20-25061

The independent factors associated with oxygen therapy COVID-19 in patients under 65 years old

PLOS ONE

Dear Dr. Liang,

Thank you for submitting your manuscript to PLOS ONE. After careful consideration, we feel that it has merit but does not fully meet PLOS ONE’s publication criteria as it currently stands. Therefore, we invite you to submit a revised version of the manuscript that addresses the points raised during the review process.

We look forward to receiving your revised manuscript.

Kind regards,

Giordano Madeddu

Academic Editor

PLOS ONE

Journal Requirements:

2. In your ethics statement in the manuscript and in the online submission form, please provide additional information about the patient records used in your retrospective study. Specifically, please ensure that you have discussed whether all data were fully anonymized before you accessed them.

Reviewers' comments:

Reviewer's Responses to Questions

**Comments to the Author**

1. Is the manuscript technically sound, and do the data support the conclusions?

Reviewer #1: Yes

Reviewer #2: Partly

2. Has the statistical analysis been performed appropriately and rigorously? 

Reviewer #1: No

Reviewer #2: Yes

3. Have the authors made all data underlying the findings in their manuscript fully available?

Reviewer #1: Yes

Reviewer #2: Yes

4. Is the manuscript presented in an intelligible fashion and written in standard English?

Reviewer #1: No

Reviewer #2: No

5. Review Comments to the Author

Reviewer #1: Ni Yue-Nan et al. submitted a manscript about the factors associated with the need of start oxygen therapy in people with SARS-CoV-2 infection under 65 years old. Many issues are present:

Abstract

The abstract is well structured. However, some sentences present mistakes. For example, the sentence “The proportion and number of young novel corona virus disease 2019(COVID-19) hospitalized patients might increase as many countries reopen.” should be reformulated.

833 should be written in letters.

The sentence “29.4% of them have coexisting condition.” is unclear.

“p” should be written in italic style

Many conjunctions are misused.

Introduction

The sentence “Meanwhile, most of the people older than 65 years old could continue to

stay at home” has no source.

I suggest adding some more information about the COVID-19 and its symptoms in the first part of the introduction. My suggestion is to write that COVID-19 is caused by a virus, SARS-COV-2, and its major symptoms are fever, cough, and dyspnea, and minor symptoms are alteration of the smell and taste, gastrointestinal symptoms, headache, and cutaneous manifestations. You could read and use these articles: https://doi.org/10.1002/hed.26269, https://doi.org/10.1002/hed.26204, https://doi.org/10.26355/eurrev_202007_22291, https://doi.org/10.1016/S1473-3099(20)30402-3, https://doi.org/10.1016/S0140-6736(20)30183-5, https://doi.org/10.1111/jdv.16669).

Methods

Which test to evaluate the probability distribution have you used? Please, add this information

The authors wrote that oxygen therapy was prescribed only when emergency signs were present. Did 63.1% of your patients have one of those emergency signs?. Please comment.

Results

The authors said “The proportion of current alchohol abuse patients was equal between the two groups (9.7% vs. 10.7%, p=0.626)”. There is a point percentage of difference, so the proportion is not equal. Please fix it, and correct the word alcohol.

In the methods, the authors wrote that fever was defined as the body temperature ≥ of 37.3°C. In the results, they wrote 37.5. Please verify which one is correct.

In the “Laboratory indices” subsection, all the Unit of measurement are missing. Please add them.

The Neutrophils / Lymphocytes ratio results to be 4.85± 6.89. A ratio can not be negative. The same goes for ALT, AST, D-dimer, urea, creatinine, and glucose. Probably you should use the median and not the mean. Please verify the distribution of all laboratory tests and use mean or median as appropriate.

The authors reported that 15 patients in the oxygen therapy group died. No data about deaths in the other group are present. Please add it.

Do the authors know the length of hospitalization for the two groups? There is a difference? Please comment.

Discussion

The author wrote, “The successful rate of oxygen therapy is high and the risk for comorbidities is low.”; it is not clear what “the risk for comorbidities is low” means. Please reformulate the sentence.

Tables

In table one, in the section about symptoms and comorbidities, the authors wrote “yes or no” in all symptoms. It is not clear what “yes or no” means. The authors should report in the subtitle which tests they used to calculate the p-value for each variable.

The Neutrophils / Lymphocytes ratio results to be 4.85± 6.89. A ratio can not be negative. The same goes for ALT, AST, D-dimer, urea, creatinine, and glucose. Probably you should use the median and not mean. Please verify the distribution of all laboratory tests and use mean or median as appropriate. For the urea, creatinine, glucose, please correct the Unit of measurement: mMol/L.

Furthermore, I suggest adding a column with the characteristics of the whole cohort.

About table 2, why urea was included in the multivariate analysis if the p-value was 0.127? Please comment.

Table 3: it is not clear what “yes or no” means.

General comments

There are many English mistakes about the use of prepositions, conjunctions, and tense. I believe that a Native speaker should revise the manuscript.

All number ≤ 10 should be written in letters. All numbers at the sentence beginning should be written in letters (example, “833 patients”, and “526” in the results section).

All abbreviations should be written entirely in the first appearance in the text. Please check all of them.

Reviewer #2: Dear Authors,

I would like to thank you for the paper you submitted and I had the chance to review. I found quite interesting the main background of the study – the importance to have data on <65 years/old COVID-19 patients to relocate resources now that younger people will be more exposed to SARS-CoV-2 for reopening of public services. The study you submitted is well-designed and inclusion criteria well-established on basis of literature, but I have a major concern I will explain as follow: in “methods” paragraph you say that the need for oxygen therapy was determined by “emergency signs” clinically established, with no mention to arterial blood gases parameters. I would like you to further explain this peculiar decision, because I found incorrect to diagnose lung failure or hypoxemia without ABG. Many recent works found properly an independent association between lower P/F ratio (Horowitz ratio) at the admission and prolonged hospitalization.

I have a minor concern about association with laboratory findings, because many data were missing for many of them and such discrepancy may have affected the solidity of statistical association. However, independent association with complications and increased death rate and lower lymphocyte count retraces experiences described in other countries, such as you may read here 10.26355/eurrev_202007_22291. Moreover, some of the blood count alterations you identified are peculiar for COVID-19 rather than other pneumonia, as you may read here https://doi.org/10.3855/jidc.12879.

In the end, if chest distress and dyspnoea are more obviously associated with respiratory failure, association with fever is less understandable. I would like you to expand more this peculiar finding, especially because no significant difference on incidence of fever was found between the two patients groups.

The language used is generally appropriate and the paper is written in acceptable English, with few typos and many grammar mistakes I suggest you to check.

6. PLOS authors have the option to publish the peer review history of their article (what does this mean?). If published, this will include your full peer review and any attached files.

Reviewer #1: No

Reviewer #2: No

---

## [Author Response · Author response to Decision Letter 0]

29 Oct 2020

Dear Editor,

Re: Manuscript ID PONE-D-20-25061

We would like to thank plos one for giving us the opportunity to revise our manuscript.

We thank the reviewers for their careful reading and thoughtful comments on previous draft.

We have carefully taken their comments into consideration in preparing our revision, which has resulted in a paper that is clearer, more compelling, and broader. The following summarizes how we responded to reviewer comments.

Thanks for all help.

Best wishes,

Dr. Liang Zongan

Corresponding Author

Review Comments to the Author 

Reviewer #1: Ni Yue-Nan et al. submitted a manscript about the factors associated with the need of start oxygen therapy in people with SARS-CoV-2 infection under 65 years old. Many issues are present:

Abstract

The abstract is well structured. However, some sentences present mistakes. For example, the sentence “The proportion and number of young novel corona virus disease 2019(COVID-19) hospitalized patients might increase as many countries reopen.” should be reformulated.

833 should be written in letters.

The sentence “29.4% of them have coexisting condition.” is unclear.

“p” should be written in italic style

Many conjunctions are misused.

Answer：

Thanks a lot for your careful reading. We have revised our manuscript item by item. 

Introduction

The sentence “Meanwhile, most of the people older than 65 years old could continue to stay at home” has no source.

Answer: 

Thanks for your advice. We have restructured this sentence and cited the resources（https://www.cdc.gov/mmwr/volumes/69/wr/mm6939e4.htm?s_cid=mm6939e4_x）

I suggest adding some more information about the COVID-19 and its symptoms in the first part of the introduction. My suggestion is to write that COVID-19 is caused by a virus, SARS-COV-2, and its major symptoms are fever, cough, and dyspnea, and minor symptoms are alteration of the smell and taste, gastrointestinal symptoms, headache, and cutaneous manifestations. You could read and use these articles: https://doi.org/10.1002/hed.26269, https://doi.org/10.1002/hed.26204, https://doi.org/10.26355/eurrev_202007_22291, https://doi.org/10.1016/S1473-3099(20)30402-3, https://doi.org/10.1016/S0140-6736(20)30183-5, https://doi.org/10.1111/jdv.16669).

Answer:

Thanks a lot for your advices. We have added more information about the COVID-19 and cited these articles. 

Methods

Which test to evaluate the probability distribution have you used? Please, add this information

Answer:

Thanks for your meaningful suggestion. We used Z test to test the distribution, and these details have been added in our manuscript. 

The authors wrote that oxygen therapy was prescribed only when emergency signs were present. Did 63.1% of your patients have one of those emergency signs?. Please comment.

Answer：

Thanks for your careful reading. Our clinical practice strictly followed the guidelines which indicated that: when emergency signs (obstructed or absent breathing, severe respiratory distress, central cyanosis, shock, coma, or convulsions) was observed, oxygen therapy should be administrated. However, due to the study design of our study, all the data was retrospectively extracted from the medical records, we could not sure whether all the patients received oxygen therapy due to they have one or more emergency signs. We have strengthened this in the limitation part of our study

Results

The authors said “The proportion of current alchohol abuse patients was equal between the two groups (9.7% vs. 10.7%, p=0.626)”. There is a point percentage of difference, so the proportion is not equal. Please fix it, and correct the word alcohol.

Answer: 

Sorry for our confusing convey. We have revised the sentence as the proportion of current alcohol abuse patients was not significantly different between the two groups (9.7% vs. 10.7%, p=0.626).

In the methods, the authors wrote that fever was defined as the body temperature ≥ of 37.3°C. In the results, they wrote 37.5. Please verify which one is correct.

Answer:

Thanks a lot for your careful reading. And sorry for our typo. It is 37.5 °C. We have revised relevant sentence in our manuscript. 

In the “Laboratory indices” subsection, all the Unit of measurement are missing. Please add them.

The Neutrophils / Lymphocytes ratio results to be 4.85± 6.89. A ratio can not be negative. The same goes for ALT, AST, D-dimer, urea, creatinine, and glucose. Probably you should use the median and not the mean. Please verify the distribution of all laboratory tests and use mean or median as appropriate.

Answer：

Thanks a lot for your careful reading. We have retested whether these variables and used median instead of mean for those without standard normal distribution. 

The authors reported that 15 patients in the oxygen therapy group died. No data about deaths in the other group are present. Please add it.

Answer:

Thanks a lot for your careful reading. No death in other group. We have added it. 

Do the authors know the length of hospitalization for the two groups? There is a difference? Please comment.

Answer:

Thanks for your meaningful advice. We have calculated the length of hospitalization for the two groups. The ones who have received oxygen therapy have longer hospitalization (16.00 vs. 14.50, p=0.002). We have added discussion about the difference in length of hospitalization between the two groups. 

Discussion

The author wrote, “The successful rate of oxygen therapy is high and the risk for comorbidities is low.”; it is not clear what “the risk for comorbidities is low” means. Please reformulate the sentence.

Answer：

Sorry for our confusing convey. We means that the risk for young Covid 19 patients developing complications during hospitalization is relatively low. We have revised our manuscript. 

Tables

In table one, in the section about symptoms and comorbidities, the authors wrote “yes or no” in all symptoms. It is not clear what “yes or no” means. The authors should report in the subtitle which tests they used to calculate the p-value for each variable.

The Neutrophils / Lymphocytes ratio results to be 4.85± 6.89. A ratio can not be negative. The same goes for ALT, AST, D-dimer, urea, creatinine, and glucose. Probably you should use the median and not mean. Please verify the distribution of all laboratory tests and use mean or median as appropriate. For the urea, creatinine, glucose, please correct the Unit of measurement: mMol/L.

Furthermore, I suggest adding a column with the characteristics of the whole cohort.

About table 2, why urea was included in the multivariate analysis if the p-value was 0.127? Please comment.

Answer: 

Sorry for our carelessness. 

① We have revised changed the convey “yes or no”.

② We have added the relevant information about which test we have used in the Method part and indicated in the Table 1 which test was used.

③ We have retested whether these variables and used median instead of mean for those without standard normal distribution. 

④ We have added the column with the characteristics of the whole cohort.

⑤ We have deleted the “Urea” in the multiple logistic regression and revised the table 2. Fortunately, all the results remained unchanged.

Table 3: it is not clear what “yes or no” means.

Answer：

Sorry for our confusing expression. We have changed the way of conveying. Hope it is clear this time. 

General comments

There are many English mistakes about the use of prepositions, conjunctions, and tense. I believe that a Native speaker should revise the manuscript.

Answer:

Thanks for your meaningful advice. We have turn to American Journal Expert, which is a professional English editing company, for help. They have helped us editing our language. 

All number ≤ 10 should be written in letters. All numbers at the sentence beginning should be written in letters (example, “833 patients”, and “526” in the results section).

Answer:

Thanks a lot for your careful reading. We have revised all the relevant numbers. 

All abbreviations should be written entirely in the first appearance in the text. Please check all of them.

Answer: 

Thanks a lot for your advice. We have checked again and revised our manuscript.

Reviewer #2: Dear Authors,

I would like to thank you for the paper you submitted and I had the chance to review. I found quite interesting the main background of the study – the importance to have data on <65 years/old COVID-19 patients to relocate resources now that younger people will be more exposed to SARS-CoV-2 for reopening of public services. The study you submitted is well-designed and inclusion criteria well-established on basis of literature, but I have a major concern I will explain as follow: in “methods” paragraph you say that the need for oxygen therapy was determined by “emergency signs” clinically established, with no mention to arterial blood gases parameters. I would like you to further explain this peculiar decision, because I found incorrect to diagnose lung failure or hypoxemia without ABG. Many recent works found properly an independent association between lower P/F ratio (Horowitz ratio) at the admission and prolonged hospitalization.

Answer: 

Thanks a lot for your careful reading and meaningful advice. We totally agree with you that without ABG, clinicians can not diagnose hypoxemia with a 100% correct rate.

As mentioned above, our study was retrospective study, we could not sure the reason for oxygen therapy in every patients, although our clinicians strictly followed the guideline published by the WHO. It is not mentioned that ABG is a remark or indication for oxygen therapy in the guideline. We guess the reason was that：since the strong infectiousness and extraordinary quick development of the disease, it is unlikely to do the ABG for every patients before oxygen therapy. Obviously, in the special pandemic, ABG is not very convenience and feasible as usual. Moreover, lack of clinicians during the pandemic may be another reason. 

However, we think your concern was totally right and reasonable. We have strengthened it in the limitation part of our manuscript. 

I have a minor concern about association with laboratory findings, because many data were missing for many of them and such discrepancy may have affected the solidity of statistical association. However, independent association with complications and increased death rate and lower lymphocyte count retraces experiences described in other countries, such as you may read here 10.26355/eurrev_202007_22291. Moreover, some of the blood count alterations you identified are peculiar for COVID-19 rather than other pneumonia, as you may read here https://doi.org/10.3855/jidc.12879.

Answer:

Thanks a lot for your meaningful advice. We should admit that the missing of laboratory findings will definitely affect our results. The results of our study have something in common with the previous published papers such as the two listed. However, the population of our study was special, younger, and otherwise, we have larger sample size. 

We have added the discussion about the differences between our laboratory findings and other published papers, and also the influence of missing laboratory data in the limitation part. 

In the end, if chest distress and dyspnoea are more obviously associated with respiratory failure, association with fever is less understandable. I would like you to expand more this peculiar finding, especially because no significant difference on incidence of fever was found between the two patients groups.

Answer:

Thanks a lot for your meaningful advice. We have expanded the discussion about fever in our manuscript. 

The language used is generally appropriate and the paper is written in acceptable English, with few typos and many grammar mistakes I suggest you to check.

Answer:

Thanks for your meaningful advice. We have turn to American Journal Expert, which is a professional English editing company, for help. They have helped us editing our language.

---

## [Decision Letter · Decision Letter 1]

24 Nov 2020

PONE-D-20-25061R1

The independent factors associated with oxygen therapy COVID-19 in patients under 65 years old

PLOS ONE

Dear Dr. Liang,

Thank you for submitting your manuscript to PLOS ONE. After careful consideration, we feel that it has merit but does not fully meet PLOS ONE’s publication criteria as it currently stands. Therefore, we invite you to submit a revised version of the manuscript that addresses the points raised during the review process.

We look forward to receiving your revised manuscript.

Kind regards,

Giordano Madeddu

Academic Editor

PLOS ONE

Reviewers' comments:

Reviewer's Responses to Questions

**Comments to the Author**

1. If the authors have adequately addressed your comments raised in a previous round of review and you feel that this manuscript is now acceptable for publication, you may indicate that here to bypass the “Comments to the Author” section, enter your conflict of interest statement in the “Confidential to Editor” section, and submit your "Accept" recommendation.

Reviewer #1: All comments have been addressed

Reviewer #2: All comments have been addressed

2. Is the manuscript technically sound, and do the data support the conclusions?

Reviewer #1: Yes

Reviewer #2: Yes

3. Has the statistical analysis been performed appropriately and rigorously? 

Reviewer #1: Yes

Reviewer #2: Yes

4. Have the authors made all data underlying the findings in their manuscript fully available?

Reviewer #1: Yes

Reviewer #2: Yes

5. Is the manuscript presented in an intelligible fashion and written in standard English?

Reviewer #1: Yes

Reviewer #2: Yes

6. Review Comments to the Author

Reviewer #1: I have reread the manuscript carefully. The authors have modified the manuscript as suggested. However, some issues are still present.

The sentence in the “Symptoms and signs at admission” paragraph is hard to read; I suggest modifying it specified which values are from the oxygen group and the other group.

Furthermore, the authors wrote, “A higher respiratory rate of 20 (19-21) vs. 20 (19-20) times/minute (p=0.005)”; I believe there is a typo in the number because it is odds that the number is equal, but the p-value is significative; then, I suggest to specify the group.

Also, in the laboratory findings, it is unclear which values are from one group and another. I suggest to specified, maybe at the start of the paragraph.

A recent study (https://doi.org/10.1111/eci.13427) found that De Ritis ratio was a predictor of hospital mortality in COVID-19 patients. In your paper, there was a significant difference regarding ALT and AST values in the two groups. Furthermore, in univariate analysis, they showed a high OR, not confirmed in the multivariate analysis. Could you verify if the De Ritis ratio was higher in the oxygen group? I believe that it could improve the value of your work.

Are you sure that the urea level (4.61±4.34) has a normal distribution? Please verify it.

It is not clear if it is a mean or a median regarding the length of hospitalization. Please specified it, and add the standard deviation or IQR, as appropriate.

In table 1, in the variable column, regarding the non-continuous variable (e.g. gender, smoker), the authors wrote that they are expressed in “no./total no. (%)”, However, the denominator is missing. In table 3, instead, it is present. Please level out this aspect.

Table 2. In the column of the multivariate p-value, the p-value is expressed without the zero before the dot. Please add the zero to level out the column with the manuscript. Furthermore, “p” in table 1 is written in capital letters. Please modify it; it should be written in lowercase and in italic style.

Besides, you should add, both in “Univariable” and in “Multivariable”, “ OR (95%CI), and explain the meaning of these abbreviations in the subtitle.

I suggest modifying “T”, with “Body temperature”.

Table 3. I suggest removing “all” and “N=526”. I suggest modifying the caption in “Treatment details and complications in 526 patients received oxygen therapy”. Furthermore, in the caption the explanation of ARDS is missing.

Reviewer #2: The authors have adequately addressed my suggestions, The article is now clear and complete and also adds new knowledge to a very little known scientific topic.

7. PLOS authors have the option to publish the peer review history of their article (what does this mean?). If published, this will include your full peer review and any attached files.

Reviewer #1: No

Reviewer #2: No

---

## [Author Response · Author response to Decision Letter 1]

27 Dec 2020

Dear Editor,

Re: Manuscript ID PONE-D-20-25061

We would like to thank plos one for giving us the opportunity to revise our manuscript.

We thank the reviewers for their careful reading and thoughtful comments on previous draft.

We have carefully taken their comments into consideration in preparing our revision, which has resulted in a paper that is clearer, more compelling, and broader. The following summarizes how we responded to reviewer comments.

Thanks for all help.

Best wishes,

Dr. Liang Zongan

Corresponding Author

Review Comments to the Author

Reviewer #1: I have reread the manuscript carefully. The authors have modified the manuscript as suggested. However, some issues are still present.

The sentence in the “Symptoms and signs at admission” paragraph is hard to read; I suggest modifying it specified which values are from the oxygen group and the other group.

Answer：

Thank you for your advice. We have revised it. 

Furthermore, the authors wrote, “A higher respiratory rate of 20 (19-21) vs. 20 (19-20) times/minute (p=0.005)”; I believe there is a typo in the number because it is odds that the number is equal, but the p-value is significative; then, I suggest to specify the group.

Answer:

Thanks for your careful reading. We have checked. The number is right as well as the p value. Although the median and Q1 were the same in the two groups，the Q3 was obviously higher in the oxygen therapy group. Which means, the percentage of patients who have RR>20 was higher in the oxygen therapy group.(we have provide the detail in the Table 1). Because the RR was not normal distributed, we used the nonparametric test, which calculated the rank of the RR value of each patient in the two groups. Since higher percentage of the patients who had RR>20 in the oxygen therapy group was found, the mean rank in the oxygen therapy group was higher and the p-value was significative. 

Also, in the laboratory findings, it is unclear which values are from one group and another. I suggest to specified, maybe at the start of the paragraph.

Answer:

Sorry for our confusing expression. We have revised this paragraph and tried to make it more clear. 

A recent study (https://doi.org/10.1111/eci.13427) found that De Ritis ratio was a predictor of hospital mortality in COVID-19 patients. In your paper, there was a significant difference regarding ALT and AST values in the two groups. Furthermore, in univariate analysis, they showed a high OR, not confirmed in the multivariate analysis. Could you verify if the De Ritis ratio was higher in the oxygen group? I believe that it could improve the value of your work.

Answer：

Thank you for your meaningful advice. We have tested the ratio of AST/ALT, however, no significant between the two groups has been found. Moreover, we have added discussion about the ratio of AST/ALT.

Are you sure that the urea level (4.61±4.34) has a normal distribution? Please verify it.

Answer:

Sorry for our carelessness. We have revised it and relevant part in the manuscript.

It is not clear if it is a mean or a median regarding the length of hospitalization. Please specified it, and add the standard deviation or IQR, as appropriate.

Answer:

Thank you for your careful reading. We have revised our manuscript accordingly. 

In table 1, in the variable column, regarding the non-continuous variable (e.g. gender, smoker), the authors wrote that they are expressed in “no./total no. (%)”, However, the denominator is missing. In table 3, instead, it is present. Please level out this aspect.

Answer:

Thank you for your careful reading. We have uniform our expression. 

Table 2. In the column of the multivariate p-value, the p-value is expressed without the zero before the dot. Please add the zero to level out the column with the manuscript. Furthermore, “p” in table 1 is written in capital letters. Please modify it; it should be written in lowercase and in italic style.

Besides, you should add, both in “Univariable” and in “Multivariable”, “ OR (95%CI), and explain the meaning of these abbreviations in the subtitle.

Answer:

Thank you for your advice, we have revised the Tables item by item. 

I suggest modifying “T”, with “Body temperature”.

Table 3. I suggest removing “all” and “N=526”. I suggest modifying the caption in “Treatment details and complications in 526 patients received oxygen therapy”. Furthermore, in the caption the explanation of ARDS is missing.

Answer:

We have revised the Table 3 accordingly.

---

## [Editor Report · Decision Letter 2]

6 Jan 2021

The independent factors associated with oxygen therapy COVID-19 in patients under 65 years old

PONE-D-20-25061R2

Dear Dr. Liang,

We’re pleased to inform you that your manuscript has been judged scientifically suitable for publication and will be formally accepted for publication once it meets all outstanding technical requirements.

Kind regards,

Giordano Madeddu

Academic Editor

PLOS ONE

---

## [Editor Report · Acceptance letter]

11 Jan 2021

PONE-D-20-25061R2 

The independent factors associated with oxygen therapy in COVID-19 patients under 65 years old 

Dear Dr. Liang:

I'm pleased to inform you that your manuscript has been deemed suitable for publication in PLOS ONE. Congratulations! Your manuscript is now with our production department. 

Kind regards, 

on behalf of

Dr. Giordano Madeddu 

Academic Editor

PLOS ONE